# Oleosome Delivery Systems: Enhancing Stability and Therapeutic Potential of Natural Products and Xenobiotics

**DOI:** 10.3390/pharmaceutics17101303

**Published:** 2025-10-07

**Authors:** Marlon C. Mallillin III, Roi Martin B. Pajimna, Shengnan Zhao, Maryam Salami, Raimar Loebenberg, Neal M. Davies

**Affiliations:** 1Faculty of Pharmacy and Pharmaceutical Sciences, University of Alberta, Edmonton, AB T6G 2E1, Canada; mallilli@ualberta.ca (M.C.M.III); shengna2@ualberta.ca (S.Z.); msalami2@ualberta.ca (M.S.); 2Department of Pharmacy, Faculty of Pharmacy, University of Santo Tomas, Manila 1015, Philippines; 3The Graduate School, University of Santo Tomas, Manila 1015, Philippines; 4China Z. Pharmaceutical Productivity Centre, Beijing 101111, China; 5Department of Food Science, Engineering and Technology, College of Agriculture & Natural Resources, Karaj Campus, University of Tehran, Karaj 31587-77871, Iran

**Keywords:** oleosomes, lipid nanocarriers, drug delivery, natural products, xenobiotics

## Abstract

Oleosomes are submicron oil bodies of a triacylglycerol core enveloped by a phospholipid monolayer and embedded proteins, forming a naturally assembled nanocarrier with exceptional oxidative resilience, interfacial stability, and biocompatibility. Their unique architecture supports solvent-free extraction, self-emulsification, and near-complete encapsulation of highly lipophilic compounds (log P > 4), including curcumin and cannabidiol, with reported efficiencies exceeding 95%. These plant-derived droplets enhance oral bioavailability through lymphatic uptake and enable targeted delivery strategies such as magnetically guided chemotherapy, which has reduced tumor burden by approximately 70% in vivo. The review critically examines recent advances in oleosome research, spanning botanical sourcing, green extraction technologies, interfacial engineering, xenobiotic encapsulation, pharmacokinetics, and therapeutic applications across oncology, dermatology, metabolic disease, and regenerative medicine. Comparative analyses demonstrate that oleosomes rival or surpass synthetic lipid nanocarriers in encapsulation efficiency, oxidative stability, and cost efficiency while offering a sustainable, clean-label alternative. Remaining challenges, including low loading of hydrophilic drugs, allergenicity, and regulatory standardization, are addressed through emerging strategies such as hybrid oleosome–liposome systems, recombinant oleosin engineering, and stimulus-responsive coatings. These advances position oleosomes as a versatile and scalable platform with significant potential for food, cosmetic, and pharmaceutical applications.

## 1. Introduction

Oleosomes, often called plant oil bodies, are now recognized as versatile colloidal carriers for food, nutraceutical, cosmetic, and pharmaceutical products. Historically observed by polarized-light microscopy, their modern architecture is well established. Each oleosome consists of a triacylglycerol (TAG) core wrapped in a phospholipid monolayer embedded with amphipathic, hairpin-shaped oleosin proteins that provide steric and electrostatic stabilization [1]. The amphipathic nature of oleosins allows seeds to store up to 60% oil within an aqueous environment without spontaneous fusion [2]. Across diverse species, oil bodies share a conserved size window of ≈0.65–2.0 µm, with TAG comprising ~94–98 wt% of the droplet and surface constituents ~1–4 wt%; the isoelectric point typically lies between pH 5.7 and 6.6, quantitatively supporting their net negative charge at physiological pH [3,4].

In drug discovery, log P reflects lipophilicity: higher values (>4–5) indicate very lipophilic compounds with low aqueous solubility and greater risk of pharmacokinetic and safety liabilities, whereas lower values (<0–1) indicate hydrophilic compounds with poorer membrane permeation; a moderate range (~1–3) is often preferred to balance solubility and permeability [5].

Early industrial interest centered on recovering edible oil, traditionally with hexane extraction. Driven by environmental and regulatory pressures, research have shifted toward aqueous, enzyme-assisted, ultrasound- and twin-screw-press approaches that recover oil without petroleum solvents, reduce energy use, and can recover ~91–93% of soybean oil as intact oleosomes at pilot scale [6]. Oleosomes also carry natural antioxidants such as tocopherols, phytosterols, and phenolic acids, which support oxidative stability superior to many synthetic lipid nanoparticles [7].

Food formulators first exploited oleosomes as low-calorie fat replacers in spreads and dressings. Sunflower, rapeseed, and almond oleosomes now enrich plant-based milk, coffee whiteners, and bakery goods, offering creamy textures and shelf-life improvements [8]. More recently, chia and flaxseed oleosomes have delivered plant omega-3 fatty acids (FAs) in beverages, providing bioavailability comparable to fish oil emulsions [9].

Pharmaceutical research has now followed. Curcumin loads passively into rapeseed oleosomes at ~97–98% encapsulation efficiency (EE), with rapid intra-oleosome diffusion and no destabilization of droplet size over 5 days [10]. Rapeseed oleosomes have also been shown to double the systemic exposure to cannabidiol (CBD) in rats by directing absorption to the intestinal lymphatics and avoiding first-pass metabolism [11]. Magnetically responsive, antibody-targeted oleosomes further reduced tumor burden by ~70% in breast cancer xenograft models, illustrating translational potential [12].

Oleosome functionality can be tailored through botanical choice and interfacial engineering. High-oleic rapeseed and hazelnut oleosomes exhibit remarkable oxidative stability, whereas omega-3-rich chia and flaxseed variants require additional antioxidants [4]. Layer-by-layer coatings with chitosan, whey proteins, or other biopolymers further improve salt, pH, and thermal tolerance and provide controlled-release properties critical for pharmaceutical and topical oleosome formulations [13].

Peanut and soybean oleosomes carry interface proteins (notably oleosins ~15–26 kDa) that are clinically relevant allergens [14]. Proteomic epitope mapping has identified multiple linear IgE-binding epitopes on soybean and peanut oleosins, including conformational sites at the N- and C-terminal amphipathic arms [15]. Quantitatively, native oleosome dispersions typically show ζ-potentials of −20 to −30 mV (e.g., rapeseed oleosomes: −21.7 ± 1.1 mV unloaded; curcumin-loaded −26.5 ± 6.7 mV), values consistent with electrostatic stabilization but also helpful in monitoring processing-induced protein exposure that may modulate IgE binding [16]. A practical allergenicity testing panel should therefore include (i) IgE immunoblots/ELISA, (ii) basophil activation assays, (iii) processing stability tests, and (iv) cross-reactivity screening with related legumes/seeds [3].

To de-risk pharmaceutical applications, clustered regularly interspaced short palindromic repeats (CRISPR)/Cas editing of oleosin loci can delete or mutate dominant IgE-binding motifs while preserving the hydrophobic hairpin that anchors oleosins in the monolayer. Structure–function studies show that intact oleosin termini govern interfacial rheology and colloidal stability [3]; selective terminal editing offers a rational path to hypoallergenic but functional variants. Recombinant expression of engineered “low-allergen” oleosins in microbial or plant hosts could also yield pharma-grade oleosomes decoupled from allergenic seed sources, complementing generally recognized as safe (GRAS)-limited food uses with safe interfaces for drug delivery.

Many botanical sources are GRAS for food applications. Nevertheless, allergenicity must be explicitly addressed in food and non-food contexts to align with Food and Drug Administration (FDA) and European Medicines Agency (EMA) regulatory frameworks [2].

## 2. Structure and Composition of Oleosomes

Oleosomes (Figure 1) are intracellular lipid droplets ranging from 0.2 to 2 µm in [9]. Each droplet consists of a hydrophobic TAG core surrounded by a ~9 nm thick phospholipid monolayer, which is rich in phosphatidylcholine and phosphatidylserine [17]. Three families of integral proteins are embedded within this interfacial layer: oleosins, caleosins, and steroleosins. Oleosins are most abundant and key to stability: a central hydrophobic hairpin anchors in the TAG core, while charged N- and C-termini extend into the aqueous phase to provide steric hindrance and electrostatic repulsion, limiting coalescence even at high cellular oil loads [3]. This protein–phospholipid shell explains much of the colloidal stability of oleosomes. Intact oleosins create a resilient interfacial film; when their hydrophilic termini are removed, emulsifying power and stability drop sharply, highlighting the functional role of the native interface [3].

Oleosomes also carry intrinsic antioxidants that help resist lipid oxidation. For example, walnut oleosomes contain tocopherols and phytosterols, which are compounds linked to oxidative protection [4]. In coconut, the interface comprises proteins and phospholipids around a TAG core rich in lauric and myristic acids, with β-sitosterol as a principal sterol, again illustrating antioxidant-bearing membranes [1].

The native interfacial proteins and co-encapsulated antioxidants give oleosomes notable physical and chemical stability, often rivaling or exceeding engineered emulsions when interfaces are preserved [1,3]. This underpins growing interest in oleosomes as natural, “clean-label” carriers, systems formulated with recognizable ingredients and minimal added emulsifiers across food, nutraceutical, and pharmaceutical products [18].

## 3. Different Extraction Methods for Natural Plant-Derived Oleosomes

Efficient oleosome recovery must liberate the droplets while keeping the phospholipid–protein membrane intact since ruptured droplets coalesce, lose emulsifying power, and oxidize faster [6,18]. The four principal methods of extraction are summarized in Table 1.

### 3.1. Organic-Solvent (Hexane) Extraction

Conventional oil-mill processing (flake → mill → counter-current hexane wash → desolventization) maximizes bulk oil yield but strips the interfacial membrane, yielding no intact oleosomes. It is appropriate when neutral oil is the objective rather than an oleosome-rich dispersion [6].

### 3.2. Aqueous Extraction

Extraction with water or mild buffer (pH 7–9), followed by wet grinding or 50–150 MPa high-pressure homogenization (HPH) and wash–cream centrifugation, preserves droplet integrity and produces ready-to-use oil-in-water emulsions [6,19]. Drawbacks include water/centrifuge demand and effluent management.

### 3.3. Mechanical Pressing/Twin-Screw Extrusion

Low-moisture single- or twin-screw routes express an oleosome-rich cream with 40–60% lower process energy than solvent routes and minimal wastewater [6]. Shear can disrupt some droplets, and the press cake may retain 10–30% oil unless re-pressed.

### 3.4. Assisted Extractions (Enzyme- or Ultrasound-Aided)

Cell-wall-degrading enzymes (e.g., cellulase, pectinase, protease) and/or low-frequency ultrasound (≈20–40 kHz) applied prior to mild homogenization increase yields at lower shear, shorten processing times, and enhance preservation of interfacial proteins; cost of enzymes, risk of over-hydrolysis, and ultrasonic scale-up remain considerations [6,18].

### 3.5. Method Selection and Application Relevance

When bulk neutral oil is required and legacy assets are available, hexane extraction remains efficient. For solvent-free, “clean-label” dispersions of intact oleosomes destined for food systems (e.g., plant-based mayonnaises, milks), aqueous or enzyme-assisted routes are preferred [18,19]. Twin-screw pressing is attractive where minimal water use, continuous operation, and low energy are prioritized [6]. Pharmaceutical and cosmetic applications that rely on native interfacial proteins for stability or for layering strategies (e.g., chitosan or whey-protein coatings) likewise benefit from methods that minimize interfacial denaturation, easing downstream functionalization [15,18].

**Table 1 pharmaceutics-17-01303-t001:** Extraction Methods for Plant-Derived Oleosomes.

Method	Typical Procedure	Advantages	Drawbacks	Representative Yields	References
Organic-solvent extraction (hexane)	Flake → mill → counter-current hexane wash → desolventize	Very high lipid recovery; uses existing oil-mill infrastructure	Large solvent use; volatile organic-compound emissions; energy-intensive recovery; and membranes are stripped (no intact oleosomes)	>97% of total seed oil; ≈0% intact oleosomes	[6,20]
Aqueous extraction	Water/mild buffer (pH 7–9); wet grind or 50–150 MPa HPH; wash–cream centrifugation	Preserves droplet integrity; solvent-free; yields oil-in-water emulsions	High water and centrifuge throughput demands; requires effluent valorization	80–96% intact oleosomes (soybean, rapeseed, and flaxseed)	[6,20,21]
Mechanical pressing/twin-screw extrusion	Low-moisture pressing/extrusion; cream separation	Continuous; 40–60% lower energy than solvent routes; minimal wastewater	Shear-induced disruption; 10–30% residual oil in press cake unless re-pressed	~60–90% oleosome recovery (rapeseed at pilot scale)	[6,21]
Assisted extractions (enzyme- or ultrasound-aided)	Cell-wall-degrading enzymes and/or 20–40 kHz ultrasound before mild homogenization	Higher yields at lower shear, shorter processing times, reduced water demand, good preservation of membrane proteins	Enzyme cost; potential over-hydrolysis; ultrasound scale-up	Enzymes up to ~93% (sunflower/peanut); enzyme + ultrasound: ~85–90% recovery	[6,8,21]

## 4. Seed-Derived Oleosomes

The unique physicochemical properties (Table 2) of seed-derived oleosomes enhance their suitability for various pharmaceutical and personal care formulations. Rapeseed oleosomes, which are mid-sized and rich in oleic acid, exhibit strong oxidative resistance and prolonged shelf-life, making them ideal for both oral and topical applications [11]. Sunflower-derived oleosomes, despite reaching sizes up to 10 µm, contain approximately 55% linoleic acid and closely mimic the lipid profile of human sebum and enhance skin compatibility [22]. Soybean oleosomes are comparable in size but often co-extract allergenic proteins like glycinin and β-conglycinin, necessitating additional purification steps before use in parenteral formulations [23]. Oleosomes from flaxseed (Figure 2) and chia are rich in α-linolenic acid (>50%), supporting cardiometabolic health, though their susceptibility to oxidation requires antioxidant stabilization [9,24]. In contrast, safflower oleosomes are small and have the highest linoleic acid content, making them cost-effective options for high-volume moisturizing emulsions in personal care [25]. Hemp oleosomes strike a balance with an optimal ω-6:ω-3 FA ratio and moderate protein surface coverage, features that have been successfully employed for CBD encapsulation [26]. Together, this botanical diversity empowers formulators to tailor oleosome-based delivery systems for oxidative stability, sensory texture, or nutritional function while minimizing the need for synthetic surfactants.

## 5. Nut- and Fruit-Derived Oleosomes

Oleosomes derived from nuts and fruits (Table 3) offer unique functionalities not typically found in conventional oilseeds. Almond oleosomes (0.2–2 µm), for example, are encapsulated in a tightly packed phospholipid shell that slows intestinal lipolysis, making them promising oral carriers for acid-sensitive bioactives [28]. Hazelnut oleosomes are larger (3–11 µm), composed of over 80% lipid but low in interfacial proteins, which enhances their spreadability and lubricity, which are desirable traits for applications such as margarines and spreads [29]. Sea buckthorn oleosomes (0.5–20 µm) are rich in palmitoleic acid; upon digestion, their free FA content has been shown to stimulate insulin secretion from both mouse and human β-cells, suggesting antidiabetic potential [30]. Although peanut oleosomes resemble those of soybean in size, they are notable for containing allergenic oleosins. Nonetheless, their balanced lipid–protein composition supports the structural integrity of emulsion-filled gels [15]. Capsicum seed oleosomes illustrate the potential for sustainable resource utilization: modeling studies indicate that their extraction requires up to 25% less energy dissipation compared with conventional pressing and refining methods [21]. Avocado mesocarp stores oil in oleosomes with variety-specific sizes of about ~12 to ~41.5 µm and Hass oil at roughly 60% oleic [31]. Coconut droplets are polydisperse (~1–30 µm) with a prominent ~14 kDa oleosin and ζ potentials near −13 mV to −33 mV indicating electrostatic stabilization [1]. An applied study reports ~93 wt% lipid and ~5 wt% protein with ~1.35 µm spheres suitable for clean-label emulsions [32]. Walnut oleosomes average 5.1 ± 0.3 µm, are highly unsaturated at about 70% polyunsaturated FA, and remain most stable far from their isoelectric point near pH 4.4 [4].

## 6. Xenobiotic Encapsulation in Oleosomes

Oleosomes offer a versatile platform for encapsulating a wide range of xenobiotics with diverse physicochemical properties as demonstrated in Table 4. Moderately lipophilic compounds such as curcumin (log P = 3.2) and highly lipophilic agents like CBD (log P = 6.5) achieve near-complete loading efficiency, resulting in enhanced photostability and improved oral bioavailability via lymphatic transport, respectively [10,11].

Recent work with hempseed oleosomes has revealed that EE is highly ratio-dependent: when CBD-to-lipid ratios exceed 1:1 (*w*/*w*), efficiency drops significantly below the ~90–99% reported at lower ratios. Furthermore, in vitro digestion models showed notably slow CBD release (~8% free FA liberation after 120 min), consistent with the protective effect of dense oleosin–phospholipid interfaces [26].

Sildenafil citrate, a molecule of intermediate polarity with an acidic pK_a_ of 9.2, has been effectively delivered using nano-oleosomes, leading to a fourfold increase in skin deposition and showing promise for the treatment of chemotherapy-induced hand-foot syndrome [33]. The permanently charged alkaloid berberine demonstrates therapeutic potential in a vitiligo mouse model when formulated in a hyaluronate-stabilized gel-core oleosome [34]. Magnetic, antibody-conjugated oleosomes carrying the alkylating agent carmustine have achieved approximately 70% tumor reduction in breast cancer models [12]. Additionally, propranolol encapsulated in chitosan-coated oleosomes effectively eradicates *Candida* biofilms while retaining high drug entrapment [13]. These examples highlight oleosomes’ ability to deliver a spectrum of compounds from moderately polar β-blockers to extremely hydrophobic carotenoids affirming their broad utility in both nutraceutical and pharmaceutical applications.

## 7. Administration Routes of Oleosome-Based Formulations

Oleosome carriers can be engineered for virtually every major dosing route, as summarized in Table 5. In oral delivery, rapeseed oleosome emulsions stimulate chylomicron assembly and bypass first-pass metabolism, doubling CBD exposure in rats [11]. Dermal preparations exploit the lipid-rich stratum corneum: a sildenafil nano-oleosome cream quadruples skin deposition, and a hyaluronate-stabilized, gel-core berberine system provides a depot effect that restores pigmentation in a vitiligo model [33,34]. Intravenously, antibody-decorated, magnetic carmustine oleosomes shrink breast-tumor xenografts by about 70% while enabling on-demand hyperthermia [12]. Vaginal chitosan-oleogels achieve complete clearance of *Candida* biofilms and add mucosal residence [13]. Finally, peanut-oleosome reinforced-chitosan hydrogels protect curcumin through the acidic stomach and release it selectively in the intestine, attenuating inflammatory markers ex vivo [15]. These selected examples highlight how the native phospholipid–protein shell of oleosomes can be paired with route-specific coatings or triggers to enhance permeation, targeting and controlled release without synthetic surfactants or harsh processing techniques.

## 8. Therapeutic Applications of Oleosome Systems

Oleosome-based systems demonstrate broad therapeutic potential across diverse clinical fields as shown in Table 6. In oncology, carmustine-loaded oleosomes guided by magnetic or antibody targeting deliver both drug and thermal energy directly to tumors, reducing xenograft size by approximately 70% [12]. Dermatological uses include the controlled release of berberine to restore pigmentation in vitiligo and nano-oleosome creams that relieve chemotherapy-induced hand-foot syndrome [33,34]. In metabolic disorders, digested sea-buckthorn oleosomes release palmitoleic acid and other FAs that enhance calcium-mediated insulin secretion [30]. Regenerative medicine also benefits from oleosome-conjugated fibroblast growth factor-1, which promotes granulation tissue formation and neovascularization, leading to nearly complete healing of full-thickness wounds [37]. Finally, safflower-derived oleosomes enable sun protection factor (SPF) 30 sunscreen formulations with an 80% reduction in UV-filter content, supporting clean-beauty and reef-safe standards [25].

## 9. Comparison of Oleosomes and Other Lipid Nanocarriers

Oleosomes, lipid droplets derived directly from plants, possess intrinsic stability and protective architecture thanks to their oleosin-protein–stabilized phospholipid monolayer. This natural design confers benefits such as self-emulsification, antioxidant effects, and sustained release of lipophilic actives (e.g., curcumin, vitamins) without artificial surfactants or solvents. Furthermore, their intact lipid–protein interface slows hydrolysis during digestion, making them especially attractive for oral delivery of lipophilic compounds [38,39,40]. Recent studies report EEs above 90–98% for highly lipophilic actives such as curcumin and CBD, with storage trials showing peroxide-induction times of >120 days at 25 °C due to the presence of native tocopherols and oleosins [7,17,21]. Industrial-scale estimates suggest oleosome concentrates can be produced at ~USD 15–20/kg, depending on source seed and purification intensity, which is considerably lower than the purified phospholipids used in liposome manufacture [6].

Conversely, synthetic carriers, including solid lipid nanoparticles (SLNs), nanostructured-lipid carriers (NLCs), and liposomes, offer precise control over formulation parameters (size, release kinetics, targeting moieties), supporting their mature status in clinical pipelines [41,42]. For example, SLNs typically achieve 60–85% EE for lipophilic drugs and <25% for hydrophiles, but oxidative degradation and polymorphic transitions can limit storage stability, with peroxide values increasing 2–3 fold after 60 days at 25 °C [43,44]. Production costs are higher, with reported industrial-scale estimates of ~USD 80–100/kg due to HPH (>500 bar) and surfactant cocktails.

NLCs improve upon SLNs by incorporating liquid lipids that enhance drug-loading capacity to 80–95%, yet oil separation above 30 °C limits shelf-life [45,46]. Liposomes can encapsulate hydrophilic cargo in the aqueous core and hydrophobic cargo within the lipid bilayer; co-encapsulation in the same vesicle is feasible, although partitioning depends on drug polarity, bilayer composition, and loading method, and very hydrophobic molecules may require specialized strategies or tend to aggregate. Liposomes provide flexibility, with EE ranging 60–95% for lipophiles and ~100% for remote-loaded amphipathic drugs [47,48,49,50], but they remain prone to oxidation and hydrolysis, typically requiring cold storage (≤4 °C) or lyophilization with cryoprotectants. Industrial production is among the most expensive, often exceeding USD 150–200/kg, due to the need for purified phospholipids and microfluidic or thin-film hydration processes [51].

Compared to these synthetic carriers, oleosomes combine high natural encapsulation, long oxidative shelf-life, and lower production costs, making them a compelling green alternative. However, their size heterogeneity and susceptibility to aggregation still necessitate stabilization strategies such as surface coating, homogenization, or spray-drying [14,52]. Table 7 summarizes the comparative performance of oleosomes and other lipid nanocarriers with qualitative and quantitative parameters, while Figure 3 illustrates the comparative structures of oleosomes and other lipid-based nanocarriers.

Emerging research suggests promising hybrid approaches, such as oleosin-coated liposomes, which combine natural protein interfaces with an engineered lipid vesicle [54]. These hybrids demonstrate improved dispersion and sustained release in vitro, although immunogenicity remains a key concern.

## 10. ADMET Advantages of Natural Oleosomes

### 10.1. Absorption

Natural phospholipid–protein interfaces and seed-FA profiles support uptake across skin and gut. Oral rapeseed oleosomes increased CBD area under the curve (AUC), C_max_, and lymph targeting in rats versus oil or emulsions, indicating enhanced enteric absorption and lymphatic routing [11]. Topically, sildenafil-loaded oleosomes boosted human skin permeation and deposition ex vivo compared with the suspension [33]. Coconut oleosomes show polydisperse droplets with negative zeta potential and robust interfacial proteins, consistent with stable dermal delivery systems [1].

### 10.2. Distribution

Strong interfacial activity and membrane integrity help maintain colloidal stability in biological fluids, which can prolong residence at target epithelia. Sunflower oleosomes act as effective interfacial stabilizers, with size-dependent intact-particle or membrane-fragment behavior that supports dispersion stability [22].

### 10.3. Metabolism

Co-extracted antioxidants and the low-oxidation state of native droplets can limit premature oxidative degradation of both carrier lipids and actives. In sunflower systems, phenolic co-extracts dominated oxidation inhibition over 120 days [7]. During digestion, processing can modulate peptide release and antioxidant capacity from soybean oleosomes, potentially affecting the metabolic milieu [23].

### 10.4. Excretion

For highly lipophilic drugs, lymphatic transport with oleosome carriers can reduce first-pass hepatic exposure, as suggested by the increased mesenteric lymph node and lymph fluid CBD levels with rapeseed oleosomes [11].

### 10.5. Toxicity

Plant-derived compositions are generally biocompatible, and topical oleosome systems showed favorable safety in skin models. Sildenafil-oleosome gels improved delivery without compromising skin integrity [33], and gel-core oleosomes for berberine demonstrated efficacy with no systemic toxicity in vivo [34].

## 11. Critical Quality Attributes (CQAs) for Oleosomes

CQAs depend on the botanical source and processing conditions, reflecting differences in droplet size and polydispersity index (PDI), zeta potential and interfacial proteins, phenolic and lipid profiles, EE, release behavior, and storage stability. Table 8 summarizes the CQAs of oleosomes.

## 12. Current Limitations of Oleosome Carriers

Despite their natural appeal, oxidative stability, and compatibility with solvent-free processing, oleosome-based carriers face several limitations that constrain their broader pharmaceutical application. First, their hydrophobic TAG core inherently favors lipophilic compounds, making them poorly suited for hydrophilic drug delivery. Water-soluble actives typically require ion-pairing, external gel matrices, or surface adsorption to achieve measurable encapsulation [17]. Even with these methods, EEs rarely exceed ~15–20%, significantly lower than the near-complete loading observed for lipophilic compounds like CBD or curcumin [10,26]. Mitigation strategies are beginning to emerge. Hybrid oleosome–liposome constructs have been suggested to overcome this barrier, leveraging the natural interfacial proteins of oleosomes with the bilayer versatility of liposomes, though robust in vivo data remain lacking. Hybrid oleosome-based platforms pair natural oleosomes with polymers or structured matrices to add mucoadhesion, mechanical strength, and improved topical delivery while preserving biocompatibility. For instance, hyaluronate gel-core oleosomes significantly enhanced skin permeation and retention of berberine while demonstrating favorable safety in vivo [34]. Beyond hybrids, bionic oleosomes—engineered to tune the core oil composition and interfacial structure—offer markedly higher payload capacity and robust stability across pH and temperature stresses [55].

Second, allergenicity remains a concern. Oleosomes derived from peanuts or soybeans often carry oleosin proteins that contain IgE-binding epitopes, raising concerns of immunogenicity in sensitive populations. Efforts to mitigate this include developing hypoallergenic fractions and recombinant oleosome shells that remove or modify immunogenic motifs [2,23].

Third, batch-to-batch variability in FA composition, antioxidant content, and residual allergens, which are driven by differences in seed cultivar and extraction conditions, poses a challenge for reproducibility. This necessitates strict raw material control and in-process analytics to align with good manufacturing practice (GMP) requirements [6,21]. Overcoming these hurdles is essential if oleosomes are to rival the pharmaceutical versatility of fully synthetic lipid-based nanocarriers.

## 13. Oleosomes for Topical and Systemic Delivery

Plant-derived oleosomes are micron-sized TAG droplets enveloped by a phospholipid–oleosin membrane that remains stable under the pH and shear conditions typically found in creams and lotions [17]. This amphiphilic shell supports cold processing and self-emulsification, eliminating the need for synthetic surfactants which is an important advantage for clean-beauty formulations [14]. Nut-based variants showcase notable sensorial benefits. For instance, hazelnut oleosomes comprising over 80% liquid lipids and less than 3% interfacial protein exhibit reduced interfacial rigidity, enhancing spreadability and lubricity in water-in-oil emulsions [29].

Safflower and sunflower oleosomes add tocopherols and phenolic compounds that slow oxidative degradation and form occlusive films that reduce transepidermal water loss. Notably, a prototype sunscreen that replaced 80% of traditional UV filters with safflower oleosomes still achieved SPF 30, underscoring their protective potential [7,25].

The thin, fluid membranes of oleosomes also facilitate controlled release. A nano-oleosome cream containing sildenafil enhanced skin deposition by 4.5-fold in a rat model of chemotherapy-induced hand-foot syndrome, significantly reducing erythema and pain [33]. Similarly, gel-core soybean oleosomes delivering berberine achieved full repigmentation in a hydroquinone-induced vitiligo model, demonstrating sustained epidermal delivery [34].

These cosmetic proof-of-concept studies have paved the way for therapeutic innovations. Orally administered rapeseed oleosomes doubled systemic CBD exposure in rats by promoting intestinal lymphatic uptake via chylomicron-mediated transport, quantified by lymph-to-plasma partition coefficients, thereby bypassing first-pass metabolism; however, no human pharmacokinetic data are yet available [11]. Intravenous injection of magnetic, antibody-targeted nano-oleosomes carrying carmustine reduced breast tumor volume by approximately 70%, demonstrating compatibility with targeted delivery and hyperthermia [12].

Advances in solvent-free, enzyme-assisted aqueous extraction now enable recovery of over 90% of intact oleosomes, while recombinant oleosin engineering allows for surface functionalization with peptides or antibodies. These innovations are bringing oleosome production closer to pharmaceutical-grade manufacturing [21]. The benefits demonstrated include enhanced texture, photoprotection, antioxidant stability, and sustained release, which position oleosomes as promising biocompatible nanocarriers for both topical and systemic drug delivery.

## 14. Oleosome as a Potential for Lymph-Directed Delivery

Oleosomes are emerging as promising vehicles for lymphatic drug delivery due to their structural and functional similarity to chylomicron precursors. These naturally occurring lipid droplets, typically ranging from 0.2 to 5 µm in diameter, encapsulate long-chain TAGs within a phospholipid monolayer stabilized by oleosin proteins. This architecture enables them to withstand the harsh conditions of gastric transit. Once in the duodenum, bile salts remodel the oleosome structure into mixed micelles, which efficiently transport FAs and co-solubilized xenobiotics into enterocytes [17]. Inside the cells, re-esterification and apolipoprotein association drive the assembly of nascent chylomicrons that subsequently enter the mesenteric lymphatic system, effectively bypassing hepatic first-pass metabolism.

Several structural features enhance the lymphatic trafficking capabilities of oleosomes. Droplet sizes below 2 µm are particularly advantageous, as they avoid entrapment in the intestinal mucus layer and fall within the optimal uptake range for enterocytes [18]. Moreover, oleosomes rich in long-chain unsaturated TAGs, such as oleic (C18:1) and linoleic acid (C18:2), are preferentially re-esterified and integrated into chylomicrons [11]. Another key advantage is their oxidative stability. Low peroxide values help preserve pancreatic lipase activity, while natural antioxidants like tocopherols and phenolics are abundant in sunflower and rapeseed oleosomes and curb lipid peroxidation [17]. Additionally, surface modifications with polymers such as pectin or chitosan can protect the oleosome during gastric passage but desorb at intestinal pH, synchronizing drug release with lipolysis [26].

From a formulation standpoint, oleosomes provide a GRAS-grade, solvent-free scaffold ideal for delivering highly lipophilic agents, including cannabinoids, fat-soluble vitamins, and immune modulators, to the lymphatic system. By carefully selecting botanical sources rich in long-chain unsaturated lipids and optimizing droplet size distribution, formulators can harness the endogenous chylomicron formation pathway to enhance systemic exposure or target lymph nodes specifically. Future research should aim to directly quantify human lymphatic uptake and investigate the co-encapsulation of synergistic antioxidants to further stabilize both the oleosome carrier and its bioactive cargo.

Food-grade oleosomes closely resemble chylomicron precursors in size, composition, and interfacial chemistry. They are sub-micron to low-micron droplets (0.2–5 µm) carrying long-chain TAGs surrounded by a phospholipid monolayer stabilized by oleosins. This architecture survives gastric transit but is rapidly remodeled by bile salts in the duodenum, yielding mixed micelles that carry FAs and co-solubilized xenobiotics into enterocytes [17]. Within the cell, re-esterification and apolipoprotein packaging divert the cargo into nascent chylomicrons that enter the mesenteric lymph, bypassing first-pass hepatic metabolism (Table 9).

## 15. Discussion

Natural plant-derived oleosomes represent a distinct class of lipid-based carriers with a naturally assembled architecture that sets them apart from conventional delivery systems like liposomes and SLNs. Unlike the bilayer vesicles of liposomes or the crystalline lipid matrices of SLNs, oleosomes feature a single phospholipid monolayer encapsulating a core of TAGs. This unique structure imparts three significant advantages with direct implications for pharmaceutical translation.

First, oleosomes enable highly efficient passive loading of hydrophobic drugs. Due to the absence of an aqueous inner phase, lipophilic compounds with log P values above 4 can readily permeate the monolayer and partition into the lipid core. EEs exceeding 95% have been reported for compounds such as curcumin and CBD [10]. Beyond lipophilic drugs, a development path exists for moderately polar/ionizable and hydrophilic cargos: (i) interfacial loading by engineering the protein–phospholipid shell to bind charged small molecules [15]; (ii) core-polarity modulation using seed-oil blends to tune solvent polarity, thereby improving partitioning of amphiphiles [56]; and (iii) hybrid architectures (oleosome–liposome or oleosome–polymer shells) to host peptides/biologics at or within the interface [54].

Second, oleosomes exhibit exceptional colloidal stability. Their interfacial proteins confer steric and electrostatic protection, eliminating the need for synthetic surfactants that often carry toxicity or regulatory concerns [17]. Preservation of intact oleosin termini is critical; proteolysis or harsh processing weakens the interfacial film and accelerates aggregation [3].

Third, oleosomes possess built-in antioxidant defenses. Tocopherols, phenolics, and phytosterols are localized at the interface, inhibiting oxidative degradation and delaying peroxide formation. As a result, oleosome emulsions exhibit oxidative stability comparable to, or even exceeding, that of refined seed oils [7].

These structural features translate into tangible performance benefits across several delivery routes. Orally administered rapeseed oleosomes, for example, engage the chylomicron pathway, facilitating lymphatic transport and can increase systemic exposure of lipophilic actives such as cannabinoids by modulating digestion and membrane density [26], while avoiding the synthetic emulsifiers found in commercial formulations [11]. Topically, oleosomes-based vehicles have demonstrated efficient delivery of UV filters at reduced active loads and attenuation of inflammatory markers in reconstructed epidermis [57]. In parenteral applications, antibody-functionalized and magnetically responsive oleosomes have been shown to combine targeted delivery and external navigation, achieving a ~70% reduction in tumor volume in breast cancer xenografts while sparing healthy tissues [12].

From a manufacturing standpoint, aqueous and enzyme-assisted extraction techniques are now on par with traditional hexane-based methods, especially when the valorization of press cake is factored into the energy calculus [21]. Innovations like membrane centrifugation and twin-screw extrusion further reduce water and energy requirements, aligning with circular-economy goals. For frozen or dried formats, manufacturers should implement controlled-rate freezing with disaccharide cryoprotectants and lyoprotectants for interfacial proteins, and verify post-thaw CQAs. For oxidation, they should combine antioxidants, metal chelators, oxygen/light-barrier packaging, and nitrogen headspace, with peroxide/thiobarbituric acid reactive substance monitoring. Reviews report spray-drying/rehydration as effective for shelf-stable oleosome powders when protected interfaces [6,18].

The differentiation between pharmaceutical and food applications is made explicit to delineate regulatory and functional requirements food uses align with GRAS/clean-label and sensory/quality priorities [18,57], whereas pharmaceutical uses require GMP, Chemistry, Manufacturing, and Controls-defined CQAs, and preclinical safety testing (including immunogenicity/complement) to advance via Investigational New Drug/Investigational Medicinal Product Dossier pathways [15,26,54].

At present, oleosome technologies are most advanced in topical/cosmetic products (e.g., sunscreen vehicles) [57]. Pharmaceutical efforts are preclinical to early translational: targeted magnetic oleosomes (oncology) [21] oleosin-coated liposomes (sustained release) [54], and gastrointestinal-tuned release/lymphatic delivery studies for cannabinoids [26].

Collectively, these features position seed- and nut-derived oleosomes as a promising, modular platform for drug delivery. By combining core-polarity engineering, interfacial functionalization, and formal immunogenicity/complement risk management, the field can extend beyond high log P actives to a broader pharmacopoeia while meeting pharmaceutical quality and regulatory expectations.

## 16. Research Gaps

Although oleosome-based delivery systems can enhance drug stability and bioavailability, significant gaps hinder their clinical translation. Most pharmacokinetic data are derived from animal studies, particularly rodents, with no human pharmacokinetic trials to date, limiting patient confirmation of absorption, distribution, metabolism, and elimination profiles [11,26]. Priority should be given to rigorously designed Phase I clinical studies that quantify oral bioavailability gains in humans and comprehensively characterize safety and tolerability. Such trials provide the definitive human pharmacokinetic evidence needed to de-risk later phases and translate promising preclinical findings into clinically actionable therapies. Comparative evidence is also sparse; while oleosomes have shown advantages over emulsions and oils, few studies benchmark them against well-established pharmaceutical carriers such as liposomes, SLNs, or polymeric systems across diverse drug classes [10,17]. At the manufacturing level, current reports are primarily confined to laboratory extraction, with limited work on quality assurance protocols to monitor droplet integrity, protein–lipid ratios, and sterility at pilot or industrial scales [6]. Moreover, techno-economic analyses remain scarce, making it difficult to assess the true cost-effectiveness of oleosome-based platforms compared to synthetic alternatives [6,58].

Equally important is the lack of systematic organ distribution and clearance studies, as only a handful of in vivo reports provide partial mapping of tissue-specific uptake [11,55]. Finally, scalability and regulatory hurdles persist. While many botanical sources are GRAS, allergenicity, particularly from peanut and soybean proteins, remains a concern, and standardized regulatory frameworks for oleosome-based pharmaceuticals have yet to be established [2].

## 17. Future Directions

Future research is converging on promising avenues to elevate oleosomes from versatile natural emulsions to precision-targeted drug carriers. One emerging direction is the application of synthetic molecular biology to engineer “designer” oleosomes. By editing the oleosin protein scaffold using CRISPR or similar genome-editing tools, it may be possible to embed cell-penetrating peptides or enzyme-cleavable motifs directly into the oleosome interface. Recombinant oleosome platforms have already been explored in proof-of-concept studies, including engineered nano-oleosomes carrying antibody-targeted chemotherapeutics. However, validation in mammalian models is still required [21].

Another critical area of development lies in stimuli-responsive coatings. Layer-by-layer assembly using biopolymers such as chitosan, pectin, or whey protein can yield oleosomes that respond to pH, enzymatic activity, or redox conditions. For example, curcumin-loaded whey–chitosan hydrogels have been shown to remain intact in gastric fluid yet release their payload in the intestine [15]. Such systems should undergo rigorous stability and bioavailability testing to ensure translational feasibility.

Expanding the botanical palette beyond mainstream oil seeds is another untapped opportunity. Underexplored species such as moringa, cupuaçu, and rambutan may reveal novel lipid profiles and bioactive micronutrients with applications in neuroprotection and photoprotection [58]. However, the stability of plant-derived bioactives must be guaranteed, supported by allergenicity assessments and careful consideration of environmental factors influencing lipid content and clinical effectiveness [2].

Regulatory standardization will also be essential to facilitate clinical translation. Establishing a pharmacopeial monograph, supported by large-animal toxicology studies that map biodistribution, complement activation, and chronic exposure risks, can accelerate clinical adoption. For drug products, clear regulatory pathways such as FDA 21 Code of Federal Regulations (CFR) 314 for New Drug Applications should be defined for oleosome-based formulations [59,60,61]. This necessitates strict compliance with current GMP standards [60], particularly in sourcing and processing field-derived raw materials.

While many plant oleosomes are GRAS for food applications, this designation does not extend to pharmaceutical contexts. Future work should explicitly define GRAS status limitations for non-food applications and align oleosome-based drug products with established FDA and EMA approval pathways [62,63,64]. This includes mandatory allergenicity testing under International Council for Harmonization of Technical Requirements for Pharmaceuticals for Human Use (ICH) S6(R1) and ICH S8 guidelines, which assess immunogenicity and hypersensitivity risks for biologics and excipients [65,66]. Ensuring that oleosome formulations undergo validated in vivo allergy testing, complemented by standardized toxicological profiling, will be crucial for regulatory clearance.

Finally, sustainability considerations remain central. Life-cycle assessments have already demonstrated that enzyme-assisted aqueous extraction from capsicum seeds reduces exergy demand by approximately 25% compared to solvent-based refining [6]. Documenting similar efficiency gains across other botanical sources could justify green certification and even carbon credit eligibility for manufacturers [21].

In summary, oleosomes represent a rare synthesis of natural abundance, molecular sophistication, and translational potential. Their future development will depend on multidisciplinary collaboration spanning plant science, colloid chemistry, pharmaceutical engineering, and regulatory science. With sustained innovation, these biologically assembled nanocarriers can be transformed from agricultural byproducts into precision therapeutics, moving seamlessly from seed to field to laboratory, through clinical trials, and ultimately to the patient.

## Figures and Tables

**Figure 1 pharmaceutics-17-01303-f001:**
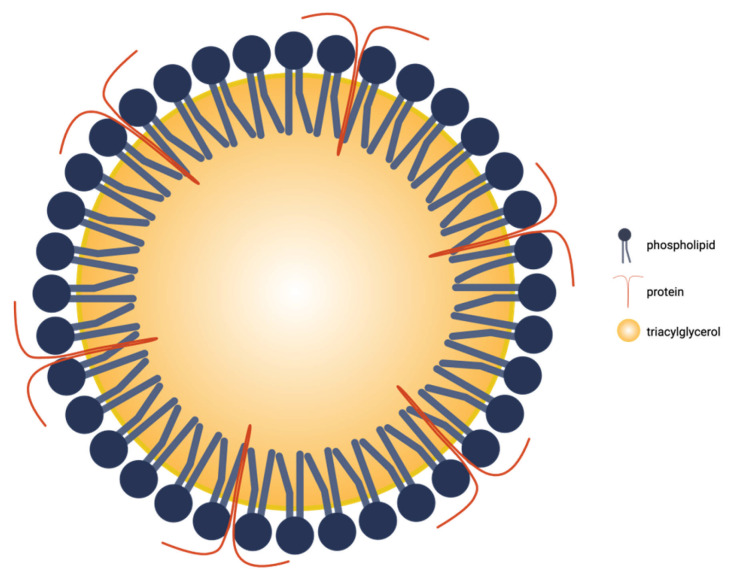
Two-dimensional schematic of a plant oleosome illustrating its TAG core, surrounding phospholipid monolayer, and surface proteins. Created in BioRender. Mallillin, M. (2025) https://BioRender.com/n51p4sw.

**Figure 2 pharmaceutics-17-01303-f002:**
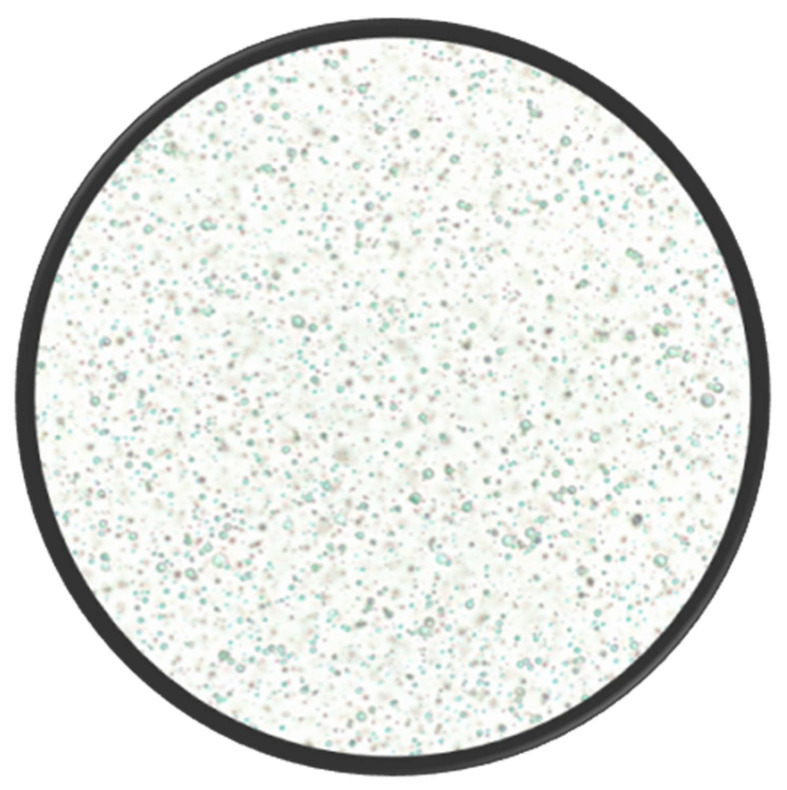
Light microphotograph of flaxseed oleosomes isolated by aqueous extraction and captured with an Echo Revolve microscope at 40× magnification.

**Figure 3 pharmaceutics-17-01303-f003:**
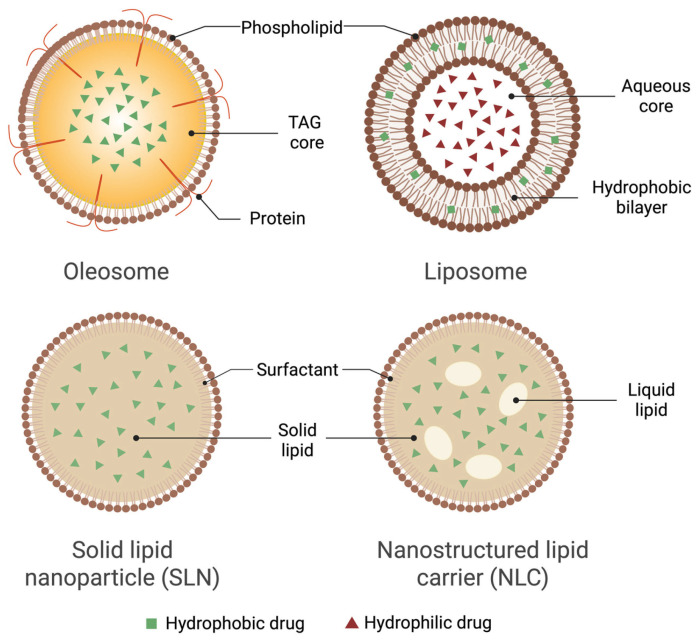
Comparative structures of lipid-based nanocarriers. Created in BioRender. Mallillin, M. (2025) https://BioRender.com/90jqj9s.

**Table 2 pharmaceutics-17-01303-t002:** Examples of Seed-Derived Oleosomes.

Seed (Species)	Size (µm)	Fat Content (wt%)	Major FA (wt%)	Protein (wt%)	Oleosome-Associated Proteins	References
Flaxseed (*Linum usitatissimum*)	0.4–2	≈6	α-Linolenic (53)	≈1	Oleosin, Caleosin, Steroleosin, Albumins/Globulins	[9]
Rapeseed (*Brassica napus*)	0.8–5	~70	Oleic (60–65)	≈2	Oleosin, Caleosin, Steroleosin	[11]
Sunflower (*Helianthus annuus*)	0.4–10	≈75–96.2	Linoleic (55)	≈3	Oleosin, Caleosin, Steroleosin, Helianthinin, Albumins	[22]
Soybean (*Glycine max*)	0.2–2	≈44.6–49.3	Linoleic (54)	≈3	Oleosin, Caleosin, Steroleosin, Glycinin, β-conglycinin	[23,27]
Chia (*Salvia hispanica*)	0.5–2	≈70–75	α-Linolenic (60)	≈1	Oleosin, Caleosin, Steroleosin	[8,24]
Safflower (*Carthamus tinctorius*)	0.5–2.5	≈75	Linoleic (70)	≈2	Oleosin, Caleosin, Steroleosin	[20,25]
Hemp (*Cannabis sativa*)	0.5–3	≈8.13–8.40	Linoleic (55)	≈2	Oleosin, Caleosin, Steroleosin	[26]

**Table 3 pharmaceutics-17-01303-t003:** Examples of Nut- and Fruit-Derived Oleosomes.

Seed (Species)	Size (µm)	Distinctive Feature	Lipid/Protein (wt%)	Reference
Coconut (*Cocos nucifera*)	~1–30	Medium-chain FA-rich core, prominent ~14 kDa oleosin, ζ ≈ −13 mV to −33 mV	93/5	[1]
Walnut (*Juglans regia*)	5.1 ± 0.3	Highly unsaturated core, strong negative ζ near neutral pH	n.d.	[4]
Peanut (*Arachis hypogaea*)	0.4–2	Allergenic oleosins reinforce gels	65/3	[15]
Capsicum (*Capsicum annuum*)	1–3	Up-cycling of spice waste, low exergy loss	67/4	[21]
Almond (*Prunus dulcis*)	0.2–2	Uniform phospholipid shell slows lipolysis	55/<3	[28]
Hazelnut (*Corylus avellana*)	3–11	High-oleic core, high lubricity	83/2.5	[29]
Sea buckthorn (*Hippophae rhamnoides*)	0.5–20	Palmitoleic-rich TAG, insulinotropic	n.d.	[30]
Avocado (*Persea americana*)	~12–41.5	Mesocarp stores oil in oleosomes, Hass ~60% oleic	n.d.	[31]

n.d. = not determined.

**Table 4 pharmaceutics-17-01303-t004:** Xenobiotic Encapsulation in Oleosomes.

Compound	Molecular Formula	MW (g mol^−1^)	log P	pK_a_	Formulation Strategy	Oleosome Source	EE (%)	Outcome	Reference
Curcumin	C_21_H_20_O_6_	368.40	3.20	9.08 (Acidic)	pH-shift diffusion	Rapeseed seed oleosomes	97.5	4.5-fold photostability	[10]
CBD	C_21_H_30_O_2_	314.50	6.50	9.13 (Acidic)	Native loading in hemp-seed oleosomes; ratio-dependent	Hemp-seed oleosomes	>90	Slow in vitro release (gradual lipid digestion); supports sustained availability	[11,26]
Sildenafil citrate	C_28_H_38_N_6_O_11_S	666.70	1.90	9.2 (Acidic); 6.5 (Basic)	Ethanol-injection nano-oleosome	Soy-phospholipid reconstituted oleosomes	95.6	4.5-fold skin deposition	[33]
Berberine	C_20_H_18_NO_4_^+^	336.40	3.60	11.73 (Basic)	Gel-core oleosome	Soy-phospholipid reconstituted oleosomes	92.3	Complete repigmentation (vitiligo model)	[34]
Carmustine	C_5_H_9_Cl_2_N_3_O_2_	214.05	1.53	13.36 (Basic)	Magnetic antibody-targeted oleosome	Engineered nano-oleosomes (olive-oil core + recombinant oleosin)	≈88	70% tumor reduction	[12]
Propranolol	C_16_H_21_NO_2_	259.34	3.48	9.53 (Basic)	Chitosan-decorated vaginal gel	Soy-lecithin oleosomes	79.6	*Candida* biofilm eradication	[13]

Data on molecular formula, molecular weight, log P, and pK_a_ were obtained from PubChem [35] and DrugBank [36].

**Table 5 pharmaceutics-17-01303-t005:** Administration Routes of Oleosome-Based Formulations.

Route	Prototype Formulation	Primary Indication	Advantage Demonstrated	Reference
Oral	CBD-loaded rapeseed oleosome emulsion	Epilepsy/chronic pain	≈2-fold higher systemic exposure through lymphatic uptake	[11]
Intravenous	Magnetically steerable carmustine oleosome	Breast cancer	Dual antibody targeting plus magnetic-hyperthermia synergy	[12]
Vaginal	Propranolol chitosan-decorated oleogel	*Candida albicans* vaginitis	Strong mucoadhesion and biofilm disruption	[13]
Oral (digestive hydrogel)	Curcumin WPI–chitosan emulsion gel	Gut inflammation/inflammatory bowel disease	pH-triggered intestinal release	[15]
Topical (dermal)	Sildenafil nano-oleosome cream	Chemotherapy-induced hand-foot syndrome	4.5-fold increase in cutaneous drug deposition	[33]
Topical (dermal)	Berberine gel-core oleosome	Vitiligo	Sustained release with accelerated repigmentation	[34]

**Table 6 pharmaceutics-17-01303-t006:** Therapeutic Applications of Oleosome Systems.

Condition	Encapsulated Cargo	Key outcome	Experimental Model	Reference
Breast cancer	Carmustine (magnetic, antibody-targeted oleosomes)	69.7% reduction in tumor volume	MDA-MB-231 xenograft (mouse)	[12]
Vitiligo	Berberine (gel-core oleosomes)	Complete repigmentation	Hydroquinone-induced mouse model	[34]
Hand-foot syndrome	Sildenafil (nano-oleosome cream)	Marked reduction in erythema and pain	Rat skin-toxicity model	[33]
Vaginal candidiasis	Propranolol (chitosan-decorated oleogel)	Total fungal clearance	Immunosuppressed rat model	[13]
Metabolic dysfunction	Sea-buckthorn free FAs (digested oleosomes)	Increased insulin secretion	MIN6 and EndoC-βH1 β-cell lines	[30]
Wound healing	FGF-1 fused oleosome protein	98% wound closure	Rat full-thickness wounds	[37]
Photoprotection	Oleosome-based UV-filter blend	SPF 30 using 80% less active	Human in vivo study	[25]

**Table 7 pharmaceutics-17-01303-t007:** Comparative Performance of Oleosomes and Other Lipid Nanocarriers.

Carrier Type	Source/Origin	EE	Stability (Shelf-Life/Peroxide Value)	Ease of Production	Approximate Industrial Cost (USD/kg)	References
Oleosomes	Natural, plant-derived droplets from seeds	90–98% for highly lipophilic drugs such as curcumin and CBD	Phospholipid–oleosin shell plus native tocopherols gives peroxide-induction times > 120 d at 25 °C	Aqueous grinding and centrifugation; no organic solvent or >500 bar HPH	15–20	[7,10,11,17,21]
SLNs	Synthetic (hydrogenated lipids + surfactants)	60–85% for lipophilic drugs; <25% for hydrophiles	Solid matrix slows leakage, but polymorphic transitions can expel drug on storage	Hot/cold HPH (>500 bar) + surfactant mix	80–100	[43,44]
NLCs	Synthetic blend of solid + liquid lipids	80–95% due to imperfect crystal lattice	More stable than SLNs yet oil separation occurs above 30 °C	Same equipment as SLNs with liquid lipid step	90–110	[45,46]
Liposomes	Synthetic/semi-synthetic phospholipid bilayers	60–95% for lipophiles; ≈100% for remote-loaded amphipathic drugs	Susceptible to oxidation/hydrolysis; often lyophilized or stored ≤ 4 °C	Thin-film hydration or microfluidics; purified phospholipids required	150–200	[47,48,49,50,53]

**Table 8 pharmaceutics-17-01303-t008:** CQAs for Oleosomes.

CQA	Importance	Primary Controls and Readouts	Reference
Droplet size and PDI	Governs uptake, skin deposition, creaming, and release	Dynamic light scattering or laser diffraction targets by route; adjust pH and shear to tighten PDI	[1]
Zeta potential and interfacial proteins	Drives colloidal stability across pH and salts	ζ at storage pH; sodium dodecyl sulfate-polyacrylamide gel electrophoresis profile of oleosin, caleosin, steroleosin	
Lipid profile and oxidative state	Impacts robustness and release of lipophilic actives	FA by gas chromatography, sterols, peroxide value, and secondary products	
Co-extracted phenolics and storage proteins	Major determinant of oxidative stability and interface behavior	Retain phenolics or add antioxidants; monitor headspace O_2_, peroxide value, aldehydes	[7]
EE and loading	Sets feasible dose and exposure	EE and loading by mass balance or ultrafiltration; pH-shift or mild co-solvent loading	[10]
Release and bioaccessibility	Links in vitro behavior to in vivo absorption	pH-stat lipolysis with micellar partitioning for oral; Franz diffusion for topical	[11]

**Table 9 pharmaceutics-17-01303-t009:** A summary of important lymphatic delivery studies.

Study	Oleosome Source/ Xenobiotic	Lymphatic Outcome	Reference
pH-stat lipolysis (in vitro) Rat in vivo pharmacokinetics	Rapeseed oleosomes/CBD	≈68% TAG hydrolysis and 90% CBD transfer into mixed micelles 2-fold higher plasma AUC and 8–26-fold greater CBD in mesenteric lymph nodes compared with bulk oil	[11]
Simulated digestion + rat in vivo	Hemp seed oleosomes/CBD	Droplets remain intact in the stomach and release CBD gradually in the intestine, supporting sustained lymph uptake	[26]
Mechanistic review	Multiple botanical sources	Identifies 0.5–2 µm diameter and unsaturated C18 TAGs as optimal for chylomicron assembly	[18]

## Data Availability

No new data were created or analyzed in this study. Data sharing is not applicable to this article.

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
