# Peer review of "Oleosome Delivery Systems: Enhancing Stability and Therapeutic Potential of Natural Products and Xenobiotics"

_pharmaceutics, 2025, doi:10.3390/pharmaceutics17101303_

Round 1
Reviewer 1 Report
Comments and Suggestions for Authors
This article reviews research on oleosomes over the past 15 years, particularly their applications in functional ingredient delivery and disease treatment, while also summarizing their limitations. The content provides guidance for future applications of oleosomes in food and medical fields. After addressing the following issues, the article is recommended for publication:
- Table 2: The total fat content of oleosomes from various sources should be provided, not just the main glyceride components and their proportions.
- Table 2: It is preferable to list the types of oleosome proteins, rather than solely reporting total protein content.
- Table 3: As a review article, the sample categories listed are too limited. Additional examples are suggested.
- Line 207: Should the statement that liposomes can simultaneously encapsulate both hydrophobic and hydrophilic materials be clarified?
- In recent years, research on recombinant oleosomes has gained prominence. These studies draw on the structure of natural oleosomes but enable targeted functional modulation. Such advancements should be included in the review to inform future research directions.
Reviewer 2 Report
Comments and Suggestions for Authors
This review sets out to take a closer look at the extraction, engineering, and applications of oleosomes, particularly focusing on their potential as sustainable nanocarriers. These nanocarriers could significantly enhance the stability, bioavailability, and therapeutic delivery of both food and pharmaceutical products. While the paper certainly piques scientific interest, there are a few important points worth mentioning:
- The restrictions in Section 10 are a bit too brief; it would be helpful to include quantitative figures regarding the inefficiencies in hydrophilic drug encapsulation (like specific EE% from existing literature) and to delve deeper into mitigation strategies, such as hybrid oleosome-liposome systems.
- There seems to be some inconsistency in Table 4 regarding pKa values and molecular formulas (for instance, carmustine's pKa is listed as both 11.96 and 13.36—let's clarify whether it's acidic or basic). Also, ensure that all log P values are properly referenced and sourced from standard databases like PubChem.
- Most of the examples are preclinical (e.g., rat models in refs [8,21]); note the lack of human trials and suggest priorities for Phase I studies to enhance bioavailability, especially of CBD-loaded oleosomes.
- In Section 9, oleosomes are qualitatively compared with SLNs/NLCs/liposomes; add quantitative parameters like mean EE%, shelf-life (e.g., peroxide values at various times), and cost per kg from industrial sources.
- In future directions (section 14), specify GRAS status limitations for non-food applications; describe FDA/EMA pathways for pharmaceuticals derived from oleosomes, such as allergy testing procedures per ICH guidelines.
- Incorporate findings from the 2025 paper on encapsulation of cannabidiol in oleosomes of hemp seed oil (PMID: 39277226), noting encapsulation efficiency decreases when CBD: lipid ratios are higher than 1:1 and slow in vitro release, to update Table 4 and discussion of pharmacokinetics.
- Describe peanut/soybean risks in section 1 with quantitative data, e.g., oleosin epitopes of proteomics studies (ref [20]), and propose CRISPR-edited hypoallergenic variants for drug application.
- For oral bioavailability (e.g., CBD doubling AUC in ref [8]), include lymphatic uptake mechanisms with schematics or partition coefficient equations, and note the absence of human data.
- In future vision, define specific FDA routes like 21 CFR 314 for oleosome-based NDAs, emphasizing the need for GMP compliance in field-sourced materials.
Reviewer 3 Report
Comments and Suggestions for Authors
This manuscript presents oleosomes as a promising natural drug delivery platform, exposes their extraordinary loading efficiency for lipophilic compounds, and demonstrates medical benefits, but suffers from a highly optimistic tone that reduces significant boundaries in hydrophilic drug distribution, manufacturing scalability, and regulatory challenges.
Title:
The title may be more brief. Consider simplifying it to increase clarity (e.g., "Olosome delivery system: enhancing the stability and efficacy of natural products and xenobiotics"). The phrase "advancing transportation" is unclear. Specifying how it moves forward (e.g., "increasing transport mechanisms") can give an accurate picture of the material.
Abstract
The initial sentence can be redesigned to simplify the details of the oleosome.
For example, you can say that "the" submicron lipid organelle is surrounded by a phospholipid monolayer made from a triglyceride core instead of being wrapped in it.
The essence can benefit from the primary purpose of review or a clear description of the hypothesis, as it currently reads more like a summary of conclusions. For clarity, briefly consider defining words like "Log P Mann," especially for the readers who may not be familiar with the vocabulary.
Introduction
While historical reference is valuable, initial comments can be streamlined to maintain reader engagement in modern applications. Consider condensing some historical details to emphasize the current relevance.
The sentence structure is sometimes complex and long, which can obstruct readability. Breaking long sentences into shorter, more digestible ones can improve clarity.
Consider defining terms like "GRAS" (Generally Recognized As Safe) at their first mention to aid readers who may not be familiar with them.
The section on allergenicity testing can be expanded to highlight the significance of these studies for potential consumer market implications.
Structure and Composition of Oleosomes
Some sentences are dense and can be simplified for better readability. For example, splitting the complex sentence regarding loss into two simpler sentences can enhance clarity.
Consider providing a clear transition between the discussion of structural components and their functional implications. This will assist the readers in making connections and highlighting the significance of each component.
The example comparing sunflower oilosome emulsion to phenolic-dilated counterparts can be more impressive if it includes specific data or references, which include quantitatively displayed studies.
The phrase "clean-label delivery vehicle" can be defined or expanded for readers who may not be familiar with the term, clarifying its relevance in food and drug contexts.
Different Extraction Methods for Natural Plant-Derived Oleosomes
A brief summary or linking statement between extraction methods and their applications will improve the flow.
Some sentences are long and may break for clarity. For example, a sentence starting with "a protein-pomppholipid shell" can be divided to increase readability. T
The details of each extraction method may benefit from a brief justification of why some methods are preferred over others, which provides reference for choices made in the field.
The section may include more specific quantitative data on yields and capacities for each extraction method, as this information will assist physicians in selecting the appropriate technique.
The references quoted within the text must be constantly formatted and ideally connected to a book list or reference section for quick access to original studies.
- Covered animal findings with diagnostic pharmacokinetics
- Use advanced imaging to imagine real-time-olosome change.
- Develop synthetic olosin to reduce variability.
Discussion
- Develop a strategy beyond lipophilic compounds (Log P > 4)
- Comprehensive immunogenicity and complementary activation studies
- Freeze with valid approaches and address oxidative issues
- Install a clear path different from food items
Research Gaps to Address:
- Direct comparison with carriers installed in many pharmaceutical classes
- Perform frequent quality checks on a pharmaceutical manufacturing scale.
- Economic viability versus synthetic options
- Systematic organ distribution and clearance data
Future Direction Refinements:
- CRISPR engineering is an exciting new area that requires a proof-of-concept study in mammals.
- The coating that responds to stimuli
- should undergo stability and biochapatability tests
- The content of the plant must be stable and undergo allergenicity testing, while recognizing the importance of environmental factors and clinical effectiveness.
Reviewer 4 Report
Comments and Suggestions for Authors
This review deals with the oleosome-based drug delivery systems. The manuscript is well-written and well-structured. A review manuscript like this is missing from the literature.
My minor comments are:
- The added value of oleosomes in the ADEMT profile of the encapsulated active substances.
- Figures and images from the literature should be incorporated in the manuscript showing drug delivery systems.
- The authors should comment on hybrid drug delivery platforms containing oleosomes.
- The CQAs of oleosomes should be added.
- The current status in the pharmaceutical industry.
Round 2
Reviewer 3 Report
Comments and Suggestions for Authors
The authors have covered all the raised issues, and the article is suitable for publication.